# Glycoprotein-Specific Polyclonal Antibodies Targeting Machupo Virus Protect Guinea Pigs against Lethal Infection

**DOI:** 10.3390/vaccines12060674

**Published:** 2024-06-18

**Authors:** Joseph W. Golden, Steven A. Kwilas, Jay W. Hooper

**Affiliations:** Department of Molecular Virology, Virology Division, United States Army Medical Research Institute of Infectious Diseases, Fort Detrick, MD 21702, USA; joseph.w.golden.civ@health.mil (J.W.G.); steven.a.kwilas.civ@health.mil (S.A.K.)

**Keywords:** New World arenavirus, Machupo virus, heterologous protection, cross-protection, antibody-based therapeutics

## Abstract

Convalescent plasma has been shown to be effective at protecting humans against severe diseases caused by New World (NW) arenaviruses, including Junin virus (JUNV) and Machupo virus (MACV). This plasma contains antibodies against the full complement of structural proteins including the nucleocapsid and envelope glycoproteins (GPcs) consisting of GP1 and GP2. To gain insights into the protective and cross-protective properties of anti-GPc-specific polyclonal antibodies, we evaluated the ability of a DNA vaccine-produced anti-GPc rabbit antisera targeting MACV strain Carvallo to provide heterologous protection against another MACV strain termed Chicava in the Hartley guinea pig model. The neutralizing activity of the rabbit antisera against the heterologous MACV strains Chicava and Mallale was found to be 54-fold and 23-fold lower, respectively, compared to the titer against the homologous MACV strain Carvallo in the PRNT50 assay. Despite lower neutralizing activity against the strain Chicava, the rabbit antisera protected 100% of the guinea pigs from this strain when administered up to four days post-infection, whereas all the control animals succumbed to the disease. Using vesicular stomatitis virus (VSV) particles pseudotyped with MACV GPc, we identified a single amino acid difference at position 122 between the strains Chicava and Carvallo GPc that significantly influenced the neutralization activity of the rabbit antisera. These findings indicate that polyclonal antibodies targeting the MACV glycoproteins can protect against lethal infection in a post-challenge setting. These data will help guide future antibody-based therapeutics development against NW arenaviruses.

## 1. Introduction

NW arenaviruses are members of the *Arenaviridae* family and are important human pathogens in the Americas [1,2]. These viruses are endemic in distinct regions due to their close association with specific rodent populations [1]. The members of this group include JUNV, MACV, and Guanrito virus (GTOV) which cause diseases in Argentina, Bolivia, and Venezuela, respectively [3,4]. JUNV is the most prominent pathogen among this group and the causative agent of Argentine hemorrhagic fever (AHF) [1,3]. Human disease ensues following exposure to the secreta and excreta of chronically infected rodent populations. NW arenaviruses infection generally results in a febrile illness that can result in vascular leakage (hemorrhage) and in severe cases lead to multi-organ failure culminating in lethal shock [5]. NW arenavirus infections can also cause neurological disease with minor symptoms including tremors or more significant symptoms including coma, encephalitis, and convulsions. The mortality rate of NW arenavirus infection is ~30% [2,6,7]. Novel human pathogenic NW arenaviruses emerge sporadically and unexpectedly [8], the most recent being a 2019 outbreak of Chapare virus in Bolivia [9] resulting in five infections, three of these fatal. Prior to this, there were other Chapare virus infections in 2003 and 2004 in Bolivia [10]. The development of countermeasures to mitigate the threat of emerging and reemerging arenaviruses thus has important public health implications.

There are no US Food and Drug Administration-licensed medical countermeasures (MCMs) that can protect humans against NW arenavirus infections in a post-exposure setting. Antibody-based products consisting of convalescent plasma taken from survivors protect infected humans against JUNV in a therapeutic setting [11,12,13]. For example, the use of convalescent plasma targeting JUNV has been shown to reduce the percentage of lethal JUNV cases from ~20% to ~1% when treatment was initiated eight days post-symptom onset [14]. Some of the plasma-treated patients developed a transient neuropathology that eventually resolved [11]. Convalescent plasma has also shown efficacy against MACV [13]. The protective efficacy of convalescent plasma is directly correlated with neutralizing antibody titers; however, some data suggest that Fc-functionality is important for protection [15].

The S-segment encodes the nucleoprotein (NP) and the glycoprotein precursor (GPC), the latter of which is cleaved into two glycoproteins GP1 and GP2 by cellular proteases [16,17]. GP1 is a surface glycoprotein and the sole target of a neutralizing antibody [18,19]. GP2 is the mediate pH-dependent viral fusion molecule [20]. Previously, our group produced anti-GPc polyclonal antibodies in rabbits using DNA vaccination and demonstrated that they protected against several NW arenaviruses, including JUNV strain Romero and GTOV strain IHD95551 in guinea pigs [21]. These findings revealed that targeting anti-glycoprotein antibodies is sufficient for protection. Our work was subsequently supported by data showing that neutralizing monoclonal antibodies targeting GP1 alone are sufficient to protect guinea pigs and non-human primates against JUNV infection [22,23]. Here, we investigated the protective efficacy of neutralizing polyclonal antibodies targeting MACV strain Carvallo GPc against an otherwise lethal infection with the MACV strain Chicava. For these studies, we used the well-established Hartley guinea pig MACV challenge system [24,25,26]. Our findings provided deeper insight into the use of antisera to cross-protect against NW arenaviruses.

## 2. Materials and Methods

### 2.1. Ethics

All the animal studies were conducted in compliance with the Animal Welfare Act and other Federal statutes and regulations relating to animals. All the experiments involving animals adhered to principles stated in the Guide for the Care and Use of Laboratory Animals, National Research Council [27]. The experimental protocols were approved by a standing internal institutional animal care and use committee (IACUC). The facilities used for this research are fully accredited by the Association for Assessment and Accreditation of Laboratory Animal Care International. The euthanasia criteria were established prior to infection studies and the animals meeting these criteria were humanely euthanized.

### 2.2. Viruses and Cells

The early passage stocks of MACV strains Chicava, Carvallo, and Mallale were propagated in Vero 76 cell monolayers (ATCC CRL-1587) as previously described [21]. A total of 293 T cells were used for all the transfection assays. The infection studies with arenaviruses were performed under BSL4 conditions.

### 2.3. DNA Vaccination

The anti-MACV strain Carvallo GPc was produced in a prior study by the DNA vaccination of rabbits [21]. Briefly, the rabbits were vaccinated three times at four-week intervals by intramuscular (i.m.) electroporation (EP) (Ichor Medical Systems Inc., San Diego, CA, USA) with 1 mg/mL plasmid DNA (pWRG/MACV-GPc) per vaccination in 0.4 mL volume. Sera collected from the animals on day 70 were used for passive transfer studies. 

### 2.4. Plaque Reduction and Neutralization Tests (PRNTs) and Virus Titer

PRNTs were performed essentially as previously described [28]. Briefly, heat-inactivated serum samples were serially diluted two-fold and combined with the indicated areanavirus to obtain ~75–100 plaques per well in a 6-well plate. Antibody/virus mixtures were incubated overnight at 4 °C and then adsorbed to confluent Vero 76 cell monolayers in 6-well plates. Following adsorption, a 2 mL solid overlay (Earle’s basal minimal essential medium (EBME), 0.5% agarose, 5% heat-inactivated FBS, and antibiotics (100 U/mL penicillin, 100 µg/mL of streptomycin, and 50 µg/mL of gentamicin) was added to each well. The plates were incubated for six days and then stained with 2 mL of solid overlay mixture that also included 5% neutral red (Gibco, Millersburg, PA, USA). The plaques were counted after overnight incubation with neutral red. Percent neutralization was calculated relative to the number of plaques in the absence of antibodies. Titers represent the reciprocal of the highest dilution resulting in a 50% reduction in the number of plaques. Data were plotted using the Graphpad Prism 6 software.

### 2.5. Plasmid Constructs

The wild-type (wt) amino acid sequence of the glycoprotein protein was derived from the NCBI reference sequence (MACV strain Chicava, accession no. AAX99327). This wt sequence was modified to generate a series of amino acid point mutations: K103R, H122Y, V185A, K212V, M231V, and D22N. These codon-optimized full-length arenavirus glycoprotein (GPc) genes were synthesized at (GeneWiz; South Plainfield, NJ, USA). The wt and point mutation genes were cloned into the NotI and BglII sites of the pWRG7077 vector and verified by sequence analysis. These arenavirus DNA constructs were used to generate pseudovirions, as previously described [29].

### 2.6. Pseudovirion Neutralization Assay (PsVNA)

The VSV pseudovirion neutralization assay (PsVNA) used to detect neutralizing antibodies in sera was described previously [29].

### 2.7. Passive Protection Studies

Female Hartley guinea pigs (300–400 g) were challenged essentially as previously described (REF). One week prior to infection, the animals were implanted with an IPTT-3000 (BMDS Inc.; Seaford, DE, USA) microchip to monitor temperature. The animals were challenged with 1000 pfu of MACV strain Chicava by the intraperitoneal (i.p.) injection of virus diluted in a total volume of 0.5 mL PBS. Antisera doses were calculated based on the equation TU/kg = (mL of antibody) × (reciprocal PRNT_80_ titers/animal weight (kg)) [14] using an average guinea pig weight of 400 g. The rabbit antiserum was administered subcutaneously (s.c.) in a total volume of 1 mL buffered with PBS at the indicated time points indicated. The antiserum doses were filtered through an ultra-low protein binding 0.22μ filter (Millipore) prior to use in the animals. The animals were weighed and monitored daily. All the challenges were conducted in a CDC-certified animal BSL4 containment facilities. The animals meeting the predetermined criteria were humanely euthanized.

### 2.8. Statistical Analysis

Survival statistics utilized the log-rank test. The statistical significance of the neutralization tests was determined using the unpaired two-tailed Student’s *t*-test. The significance of survival was determined by the log-rank test. Significant levels were set at a *p*-value less than 0.05. All analysis was performed using the GraphPad Prism 10 software.

## 3. Results

### 3.1. Polyclonal Antisera Targeting MACV Strain Carvallo Has Reduced Neutralization Titers against Heterologous Strains Chicava and Mallale

Prior to the evaluation of the rabbit antisera in the guinea pig model, the extent to which the antibodies produced against the strain Carvallo and how they could neutralize other strains of MACV was evaluated by a plaque reduction assay. The rabbit antisera were produced against the strain Carvallo in a previous study [21]. For that vaccination, four rabbits were vaccinated with pWRG/MACV-GPc (opt) three times at four-week intervals with 1 mg of plasmid DNA per vaccination using an i.m. electroporation device. Here, we found that this product neutralized the strains Chicava and Mallale less efficiently compared to the targeted strain Carvallo (Figure 1). The geometric mean 50% plaque reduction titer (PRNT50) against the Carvallo virus was 5120, but only 95 and 226 against the strains Chicava and Mallale, respectively. Thus, Carvallo-targeting rabbit sera neutralized strains Chicava and Mallale 54-fold and 23-fold lower than strain Carvallo. Both decreases in neutralization titers were statistically significant (*t*-test; *p* < 0.05). These findings indicated that polyclonal rabbit antisera targeting a GPc from a specific strain of MACV (Carvallo) has reduced neutralizing potency against heterologous strains (Chicava and Mallale).

### 3.2. Rabbit Antisera Targeting MACV Strain Carvallo GPc, Protects Guinea Pigs against Strain Chicava

MACV strain Chicava along with strain Carvallo produce a severe disease in guinea pigs [24,25,26,30]. The lethality of strain Mallale in guinea pigs is unclear; therefore, we evaluated if the strain Carvallo-targeting rabbit antibodies, despite the lower PRNT, could protect the guinea pigs against lethal disease when they were infected with the strain Chicava. Four groups of six guinea pigs were infected with MACV strain Chicava at (1000 pfu) by the intraperitoneal route (Figure 2). Anti-MACV strain Carvallo GPc-specific antibodies were administered subcutaneously to guinea pigs either once on day −1 or two doses on days 2/7 or day 4/9. A standardized method for calculating arenavirus therapeutic antibody dosage (therapeutic units; TU) was previously established by Enria et al. [14], and used in our previous study [21]. We based the antibody dose on that for strain Carvallo, 15,000 TU/kg; however, using the PRNT50 titers for Chicava, the effective dose was ~1000 TU/kg or 15-fold lower against the heterologous challenge virus. As a control, one group of guinea pigs received anti-Andes virus antisera (day −1 only) produced in the same manner as the anti-MACV antibodies. The control animals began to lose weight starting between days 2 and 15 (Figure 2A,B). They also developed a fever, peaking around day 12 (Figure 2C). All the control animals (Group 4) succumbed to the strain Chicava infection by day 24 with a mean time to death (MTD) of 22.5 days. In marked contrast, all the animals receiving antibodies against MACV survived the infection without any outward signs of disease including fever or weight change. This protection was significant for both the groups (*p* = 0.0008; log-rank) against the control group. These data showed that the polyclonal antisera could protect against heterologous MACV strains when given post-virus exposure.

### 3.3. Identification of Amino Acids That Influence Polyclonal Antibody Neutralization against MACV Strains Carvallo and Chicava

We next explored the GP1 molecular factors impacting antibody neutralization against MACV strains. There are six amino acid differences between the GP1 of the strain Carvallo and strain Chicava (Figure 3A). We examined the impact these differences had on polyclonal antibody neutralizing using VSV particles pseudotyped with GPc from Chicava, Carvallo, and the single point mutations in the GP1 molecule that changed the Chicava sequence to Carvallo. Consistent with our plaque assay data, the VSV pseudotyped neutralization test showed that antibodies neutralized Carvallo to a higher level compared to Chicava (Figure 3B). This analysis also identified a single amino acid change at position 122 (H122Y) that significantly (*t*-test; *p* < 0.05) impacted neutralizing titer. Another mutation at position 185 (V185A) increased neutralization titers, but not significantly. These data indicate that small amino acid differences in GP1 can impact polyclonal antibody neutralization.

## 4. Discussion

### 4.1. Heterogeneity in GPc Does Not Impact Protection of Polyclonal Antibodies Targeting MACV

A caveat to using glycoprotein-targeting antibodies to protect against arenaviruses is the high amount of sequence variation in the GP1 between different strains and variants. Candara et al. showed that while there are no serotypes among NW arenaviruses per se, neutralizing the antibody is generally more potent against homologous strains [31]. This structural flexibility in GP1 may be an important factor in the establishment of persistent arenavirus infections in *rodentia*. During the course of these persistent infections, antibodies produced against the parental strain used for infection poorly neutralize progeny viruses which evolve to partially evade the host immune system [32]. This natural process of GP1 modification may also contribute to the emergence of novel arenaviruses capable of infecting humans due to a gained ability to bind the human transferrin receptor 1 [33,34]. The intrinsic malleability of the NW arenavirus GP1 molecule is therefore an important feature for viral fitness and maintenance in nature. Unfortunately, this variation could degrade the efficacy of immunotherapeutic approaches against NW arenaviruses. Our findings indicate that this is not a major concern at least for antibody-based therapeutics against MACV. MACV strain Chicava and Carvallo have 98.4% GP1 amino acid similarity, and yet the neutralizing antibodies generated against Carvallo were much less potent at neutralizing Chicava. Indeed, a single amino acid at position 122 was important for strain neutralization. Amino acid 122 in the MACV strain Carvallo GP1 directly interacts with the human transferrin receptor 1 [33]. While not significant, alterations in position 185 also influenced neutralizing activity. These data exemplify the fact that minor alterations in the GP1 molecule can greatly impact even polyclonal antibody neutralization. Nevertheless, Carvallo-targeting rabbit antisera protected against the strain Chicava even when treatment was delayed four days after virus exposure and the dose of antibody used was calculated based on the PRNT values of the strain Carvallo (15,000 TU/kg), making the actual dose against the strain Chicava at least 10-fold lower. These findings suggest that GP1 heterogeneity does not dramatically degrade protective efficacy and lends further credence to the use of antibodies in a viable MCM strategy against NW arenaviruses. Furthermore, they suggest that a much lower dose of antibodies could be used to protect hosts against MACV, and possibly the other NW arenaviruses. Our findings also support the use of the heterologous strains of arenavirus in the evaluation of antibody therapeutics to verify protection against related, but not identical strains. This may be most critical in the evaluation of monoclonal antibodies.

### 4.2. Evaluation of Antibody Protection in the Hartley Guinea Pig Model

Previously, we evaluated the protective efficacy of NW arenavirus polyclonal antibodies produced by the DNA vaccination of rabbits [21]. We found that antibodies produced against JUNV and GTOV protected against severe infection when administered post-infection. With JUNV, partial protection was established when virus-specific anti-GPc antibodies were administered starting four days post-infection, but protection was lost when given on day 6. At that time, our MACV Carvallo strain was found to be attenuated [21] due to a subsequently identified mutation in the L-segment intergenic region [25]. We had demonstrated that anti-MACV rabbit polyclonal antibodies limit MACV strain Carvallo infection, but its protective efficacy against severe disease remained untested. We have subsequently identified a strain of Carvallo with an intact intergenic region that is partially lethal in guinea pigs (63% mortality); however, these studies also found that the strain Chicava was universally lethal, consistent with several other studies [24,25,30]. Therefore, here we evaluated the protection of our rabbit antisera against the strain Chicava to provide a more stringent evaluation of protective efficacy and to address key questions regarding the impacts of heterogeneity on antibody-based protection.

Other MACV animal systems exist including a STAT1-deficient murine model [35]. The STAT1 mutation greatly impairs the type I and II interferon responses in these mice making them hypersensitive to infection. However, Hartley guinea pigs are immune intact, and outbred, providing a more natural system to study infection. Accordingly, unlike STAT1-deficient mice, type I interferon is functional in these animals [36]. This may be particularly critical as NW arenavirus pathogenesis in humans and guinea pigs may be partially dependent on adverse effects of type I interferon [36,37,38,39]. Thus, the Hartley guinea pig infection model is an important platform for immunotherapeutic evaluation in an immune intact animal system.

### 4.3. Antibody-Based Therapeutics against Emerging Arenavirus Threats

Antibody-based countermeasures in the form of convalescent plasma have been used to treat NW arenavirus infections since the 1960s [11,13,14,40]. These efforts have shown great promise, for example, the initiation of treatment against JUNV within eight days of symptom onset significantly reduces mortality [11,14,40]. Similarly, the treatment of humans with convalescent plasma against MACV also appears to reduce disease severity [13], though contrary to JUNV, it is less clear. Because convalescent plasma is not a well-defined product and has significant safety concerns, recently various groups have begun to develop glycoprotein-targeting antibody therapeutics against NW arenavirus. These antibodies have been shown to be effective against authentic viruses in guinea pigs and NHPs [21,22,23]. Another group developed glycoprotein-targeting monoclonal antibodies (MAbs) against MACV and showed that they protected STAT2-deficient mice against a vesicular stomatitis virus expressing the MACV GPc [41]. Some evidence suggests that MACV GPc may be less heterogenic compared to other NW arenavirus, such as the Chapare virus [42]. Our data indicate that single amino acid changes can impact even polyclonal neutralizing activity. Such changes may be more detrimental to neutralization by single epitope targeting MAbs. These factors support the use of polyclonal antibody approaches in the development of NW arenavirus immunotherapeutics. Polyclonal antibodies would be more inherently resistant to GP1 flexibility due to their ability to target multiple epitopes. Alternatively, the combinations of MAbs targeting different epitopes may also provide the same multiepitope coverage. Previously, we demonstrated that combination DNA vaccines targeting multiple NW arenaviruses simultaneously produced neutralizing antibodies against four NW arenavirus targets (JUNV, MACV, GTOV, and SABV) [21], as well as Zika virus [43] and multiple hantaviruses [44,45]. This technology combined with human antibody expressing bovine [44,45,46] may be highly advantageous in producing an antibody-based product with broad NW arenavirus protection. Such a product could help prevent severe disease by novel arenaviruses.

## 5. Conclusions

We conclude that immune serum produced by DNA vaccination is sufficient to protect against a heterologous strain of Machupo virus in the Hartley Guinea pig model even when administered up to four days after challenge. In addition, we demonstrated that GPC amino acid 122 was import for binding of cross-neutralizing antibodies.

## Figures and Tables

**Figure 1 vaccines-12-00674-f001:**
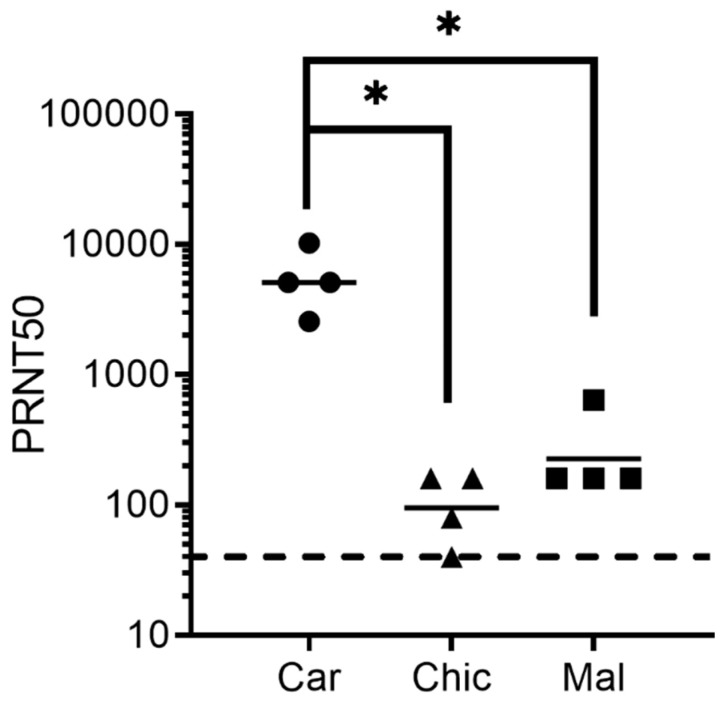
Comparison of the neutralizing activity of the anti-MACV Carvallo GPc rabbit sera against MACV strains. The PRNT50 values were determined using the serum from four rabbits vaccinated with plasmid DNA encoding the Carvallo GPc as previously reported [21]. The PRNT50 titers were determined using the MACV strains Carvallo, Chicava, and Mallale. Each data point represents a single rabbit, with the geometric mean shown by the line. The dashed line is the limit of detection for the assay. Significance is noted by the asterisks (*t*-test; *p* < 0.05).

**Figure 2 vaccines-12-00674-f002:**
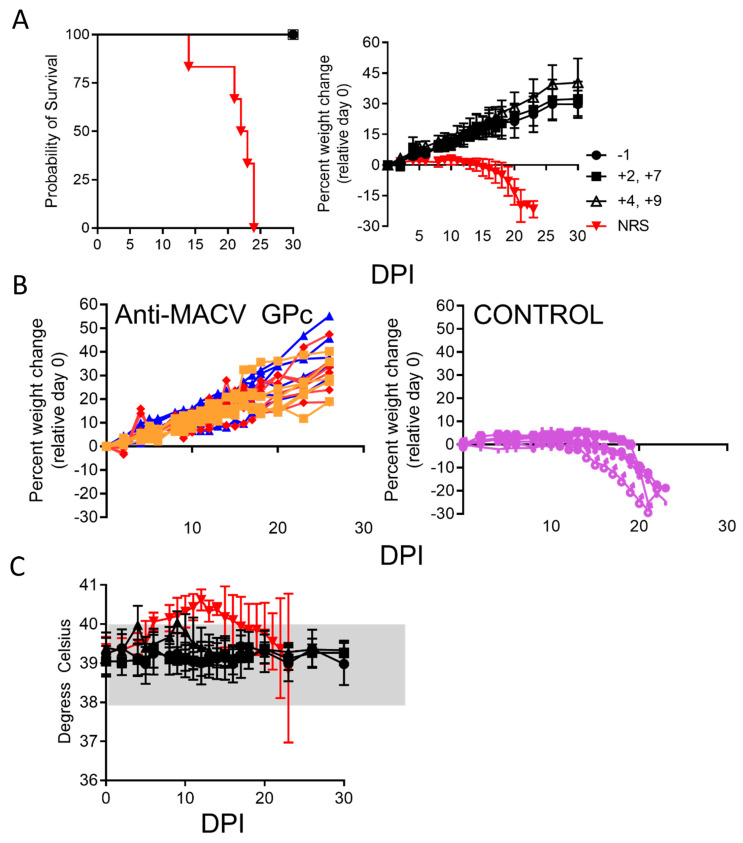
Protective efficacy of the glycoprotein-targeting antiserum against MACV strain Chicava challenge in guinea pigs. (**A**) Guinea pigs (n = 6/group) were treated with antisera (anti-MACV) or control antisera at the indicated time points. The guinea pigs were infected by the i.p. route with 1000 pfu MACV strain Chicava. The animals were injected s.c. with 15,000 TU/kg (based on a 400 g guinea pig) of rabbit anti-MACV Carvallo glycoprotein or control antiserum diluted in PBS. TU determined antibody was based on PRNT80 valuates against the strain Carvallo. A second dose of antiserum was administered five days later. Survival was monitored for 30 days post-infection. The survival and percent group weight loss change was plotted. The weights were calculated based on day 0 starting weight. (**B**) The weight changes of individual animals treated with anti-MACV GPc antiserum (left) or control antiserum (right) were plotted. (**C**) Temperatures were monitored daily by IPPT-300 implants. The mean group temperatures were plotted. The normal guinea pig temperature range is shaded in grey.

**Figure 3 vaccines-12-00674-f003:**
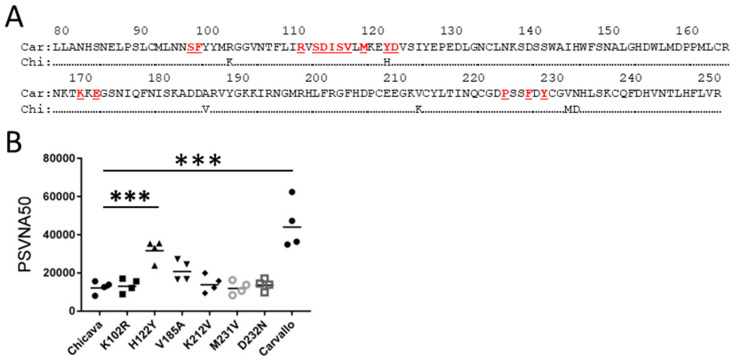
Functional mapping of the critical neutralizing antibody epitopes on the MACV GP1 molecule. (**A**) The comparison of the Carvallo and Chicava GP1 predicted amino acid sequence between regions 80 and 250. Red letters indicate the amino acids interacting with the human transferrin receptor based on crystal structure analysis [30]. (**B**) PsVNA assays were performed in triplicate with the particle pseudotyped with the indicated MACV GPc. The particles were neutralized with rabbit antiserum targeting the MACV strain Carvallo GPc produced by DNA vaccination of rabbits. Asterisks denote statistical significance (*t*-test; *p* < 0.05).

## Data Availability

The corresponding author will provide data upon request.

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
