# Peer review of "Glycoprotein-Specific Polyclonal Antibodies Targeting Machupo Virus Protect Guinea Pigs against Lethal Infection"

_vaccines, 2024, doi:10.3390/vaccines12060674_

Round 1

Reviewer 1 Report

Comments and Suggestions for Authors

Some members of New World (NW) arenaviruses, such as Junin virus (JUNV) and Machupo virus (MACV), cause lethal infections in humans. Previous studies have shown that convalescent plasma from patients infected with JUNV can partially protect patients from lethal outcomes. It should be noted, however, that approximately 10% of patients who receive convalescent plasma develop a late-onset neurological syndrome.

In this study, the authors used previously established anti-MACV Carvallo strain glycoprotein (GPc) rabbit serum to investigate its cross-neutralizing activity against other heterogenic MACV strains, Chicava and Mallale. The findings indicate that the rabbit serum displayed partial cross-neutralizing activity against these viruses. Despite showing lower neutralizing activity against the heterogenic MACV strain, administration of the rabbit serum protected Hartley guinea pigs when challenged with the heterogenic MACV Chicava strain.The authors also analyzed the amino acid sequence of MACV GP1 and identified that a variation at position 122, located in the interaction region with human transferrin receptor 1 (hTfR1), plays a crucial role in the neutralizing activity of the rabbit serum. This was determined using a pseudovirion neutralization assay (PsVNA) with vesicular stomatitis virus (VSV) pseudovirions with MACV Chicava GPc or its mutants.

This manuscript demonstrates that polyclonal anti-MACV Carvallo GPc antibodies can protect guinea pigs from lethal infections caused by heterogenic MACV strains, providing valuable insights for the development of vaccines and antiviral drugs. The scientific findings and insights presented are generally robust, with a few exceptions. However, the quality of the writing is not good enough and significant revisions are needed to get this manuscript published.

Major Comments:

1.     There is a noticeable inconsistency in terminology throughout the manuscript, which may confuse readers. For example, terms such as "rabbit serum," "polyclonal antibodies," and "antibody" are used interchangeably. Using consistent terminology would simplify the reader's understanding. Furthermore, the use and definition of abbreviations are not consistent. For instance, "glycoprotein" is abbreviated as GPc in the context of glycoprotein complexes. It would be more appropriate to consistently use "anti-GPc-antibody" instead of "anti-glycoprotein-antibody." The term "Hartley guinea pig system" should also be revised to "Hartley guinea pig model." Additionally, typographical errors throughout the document require correction after a thorough review.

2.     The authors utilize anti-MACV GPc rabbit serum in all experiments, a serum previously reported in their publications. Although references are made, this rabbit serum is a crucial material in this study. Hence, the authors should provide detailed information on how the DNA vaccine was designed, the immunization schedule for the rabbits, the serum collection timeline, and the antibody purification process.

3.     The term "Immunotherapeutics" used throughout the manuscript is not suitable for this context. It implies methods that stimulate the immune response to treat diseases, whereas the maniscript is about monoclonal and polyclonal neutralizing antibodies. "Antibody therapeutics" would be a more appropriate term for this manuscript.

Minor comments:

Line 12: “enveloped glycoproteins (GP)” should be “enveloped glycoprotein (GPc)”.

Line 13: “anti-glycoprotein-specific” should be “anti-GPc-specific”.

Line 14: Since the rabbit serum was established using a DNA vaccine with MACV Carvallo GPc gene, “anti-GP1/GP2” is better to revised to “anti-GPc”.

Line 15: “strain Chicava” should be “the other MACV strain Chicava”.

Lines 16-18: The sentence is difficult to follow. “The neutralizing activity of the rabbit antisera against heterologous MACV strains Chicava and Mallale was found to be 54-fold and 23-fold lower, respectively, compared to the titer against the homologous MACV strain Carvallo in PRNT50 assay.” might be better.

Line 18: “reduced” should be “lower”.

Line 18: “antisera” should “rabbit antisera”.

Lines 18-19: “protected 100% of guinea pigs” might be better “protected 100% of guinea pigs from lethal challenge of strain Chicava”.

Line 20: Please type out “GPc”.

Lines 21-22: The sentence might be better modified to “we identified a single amino acid difference at position 122 between strains Chicava and Carvallo GPc that significantly influenced the neutralization activity of the rabbit antisera.”.

Line 24: The term “future immunotherapeutic” looks not suitable to put this sentence. “future antibody therapeutics” might be better.

Line 26: The “immunotherapeutic” should be removed from Keywords. 

Line 29: “NW” should be “New World (NW)”.

Line 32: “Guanrito virus (GTOV)” should be “Guanarito virus (GTOV)”.

Line 38: “disease” should be “diseases”.

Lines 49-55: The authors need to mention side effects of convalescent plasma therapy causing neurological symptoms.

Line 58-59: Is GP1 the sole target of neutralizing antibodies? As for Lassa virus neutralizing antibodies, the complex of GP1 and GP2 is also target of some neutralizing antibodies. It is possible that neutralizing antibodies targeting GP2 or GP complex will be discovered, just not known so far. Therefore, it is better to use “major” instead of the word “sole”.

Line 60: “glycoprotein” should be “GPc”.

Line 59-61: This sentence lacks important information. Please include which virus and strain were used for DNA vaccination. 

Line 67: “neutralizing polyclonal antibodies targeting MACV strain Carvallo” should be “neutralizing polyclonal antibodies targeting MACV strain Carvallo GPc”.

Line 69: “immunotherapeutics” should be “antisera”.

Lines 72-73: “Polyclonal antibodies against MACV strain Carvallo has reduced neutralization titers against heterologous strains Chicava and Mallale” should be “The rabbit antisera against MACV strain Carvallo GPc has lower neutralization titers against heterologous strains Chicava and Mallale”.

Line 73: “infection studies” should be “evaluation of the rabbit antisera in the guinea pig model” 

Line 75: “plaque assay” should be “plaque reduction and neutralization tests (PRNTs)”.

Lines 75-76: Please explain how the rabbit antisera were produced.

Line 78: “PRNT50” should be “50% plaque reduction titer (PRNT50)”.

Line 78-88: In the manuscript, the PRNT50 value of the rabbit antisera against Mallale is stated as 26, but in Figure-1, it appears to be over 100 (it seems to be 222?). Please verify the data and correct it to reflect the accurate values.

Line 81: “parental” should be “homologous”.

Line 84: Please use the strain name instead of abbreviations.

Line 85: The title of Figure-1 is better to be revised. “Comparison of neutralizing activity of anti-MACV Carvallo GPc rabbit sera against MACV strains”.

Line 85: Please remove “A”.

Lines 85-88: The figure lacks sufficient explanation. It is unclear whether each point represents the results from four different serum samples or the average of four separate experiments. The authors have not provided information about the rabbit serum, so it is also unclear whether the serum was taken from individual rabbits or pooled from multiple sources. Additionally, there is no explanation provided for the asterisks used in the figure.

Line 89: “Rabbit antibody” should be “Rabbit antisera”.

Lines 110-120: The figure is missing branch labels such as A, B, and C for each element, making it difficult to identify the corresponding data. Additionally, the absence of a legend adds to the confusion about which data points represent which measurements. The lines and markers in the figure are also too thick and large, complicating the distinction between different data points. To improve clarity, it would be advisable to either separate the data into different graphs for each group or use thinner lines and smaller markers to make the distinctions clear. Additionally, the figure lacks descriptions of the statistical analyses and explanations for the asterisks. To improve clarity and validity, it's important to include details on the statistical methods used and to clarify the significance indicated by the asterisks.

Line 118: “weight loss” should be “weight change”.

Line 123, “changes” should be “differences”.

Line 134: In figure-3 A, it is advisable to use full strain names instead of abbreviations. Additionally, since the Mallale strain is also used in the experiments in Figure-1, it would be beneficial to include the amino acid sequence of GP1 from this strain as well. Furthermore, since not all amino acid sequences of GP1 were compared, it should either be specified that only amino acids 80 to 250 of GP1 are compared, or a comparison of the entire amino acid sequence of GP1 should be conducted.

Line 137: “human transferrin receptor” should be “human transferrin receptor 1”.

Line 162: “Tf1 receptor” should be “transferrin receptor 1”

Line 240: “293T cells” should be “HEK293T cells”.

Lines 242-243: The authors should explain how the rabbit antisera were obtained.

Lines 245-246: In this manuscript, the authors did not use purified antibodies.

Line 260: Please remove “5.”.

Line 261: “The wild type (wt) sequence of the glycoprotein gene” should be “The wild type (wt) amino acid sequence of the glycoprotein”.

Line 276: Vero-96 cell monolayer.

Author Response

  1. There is a noticeable inconsistency in terminology throughout the manuscript, which may confuse readers. For example, terms such as "rabbit serum," "polyclonal antibodies," and "antibody" are used interchangeably. Using consistent terminology would simplify the reader's understanding. Furthermore, the use and definition of abbreviations are not consistent. For instance, "glycoprotein" is abbreviated as GPc in the context of glycoprotein complexes. It would be more appropriate to consistently use "anti-GPc-antibody" instead of "anti-glycoprotein-antibody." The term "Hartley guinea pig system" should also be revised to "Hartley guinea pig model." Additionally, typographical errors throughout the document require correction after a thorough review.
  • Author response: We have standardized the terminology and made the changes listed below in minor comments.
  1. The authors utilize anti-MACV GPc rabbit serum in all experiments, a serum previously reported in their publications. Although references are made, this rabbit serum is a crucial material in this study. Hence, the authors should provide detailed information on how the DNA vaccine was designed, the immunization schedule for the rabbits, the serum collection timeline, and the antibody purification process.
  • Author response: The detail of the rabbit vaccinations has been added to the manuscript
  1. The term "Immunotherapeutics" used throughout the manuscript is not suitable for this context. It implies methods that stimulate the immune response to treat diseases, whereas the maniscript is about monoclonal and polyclonal neutralizing antibodies. "Antibody therapeutics" would be a more appropriate term for this manuscript.
  • Author response: These changes have been added

Minor comments: •      Author response: All of these minor edits have been made. Thank

Reviewer 2 Report

Comments and Suggestions for Authors

This study reported polyclonal antibod-22 ies targeting the MACV glycoproteins can protect against lethal infection in a post-challenge setting.

However, the submission need deeply and systemic revision.

Overall,  the English is very difficult to understand. Some sentences were hard to read, such as Lines 101-102, "As a control, one received anti-101 Andes virus antisera (day -1 only) produced in the same manner as the anti-MACV anti-102 bodies".

Lines 108-109, "These data indicated that polyclonal antibodies protect against heterologous MACV 108 strains when given post-virus exposure." This sentence was overstated and could not support by the results.

The evaluation tests were too simple, and the testing related to immune or immune cell indexes should be added.

"milliliter" and "microliter" should be "μL" and "mL", not "μl" and "ml".

Numbers and Units should be separated by a blank space.

Comments on the Quality of English Language

The English is very difficult to understand. Some sentences were hard to read.

Author Response

Reviewer #2

This study reported polyclonal antibod-22 ies targeting the MACV glycoproteins can protect against lethal infection in a post-challenge setting.

However, the submission need deeply and systemic revision.

Overall,  the English is very difficult to understand. Some sentences were hard to read, such as Lines 101-102, "As a control, one received anti-101 Andes virus antisera (day -1 only) produced in the same manner as the anti-MACV anti-102 bodies".

Lines 108-109, "These data indicated that polyclonal antibodies protect against heterologous MACV 108 strains when given post-virus exposure." This sentence was overstated and could not support by the results.

  • Author response: We do believe the data do support this statement. Antibody was given several days post-infection and the guinea pigs survived an otherwise lethal infection. The control animals did not survive. The challenge stain was heterologous (strain Chicva) as the antiserum was produced against strain Carvallo. The only interpretation is that the animals were protected by an otherwise lethal infection due to the antibody-based product. We now added statistical data to support the conclusion, all anti-MACV GPc antibody treated groups were significantly protected from lethality compared to the control animals, for both groups the p value was 0.0008.

The evaluation tests were too simple, and the testing related to immune or immune cell indexes should be added.

  • Author response: Our evaluation was survival against an otherwise lethal infection. This is the industry standard. The exact mechanism of antibody-protection (i.e. virus neutralization, Fc-mediated) is a matter of future study and beyond the scope of this current initial study.

"milliliter" and "microliter" should be "μL" and "mL", not "μl" and "ml".

  • Author response: This change was made

Numbers and Units should be separated by a blank space.

Comments on the Quality of English Language

The English is very difficult to understand. Some sentences were hard to read.

  • Author response: We have re-edited the manuscript and sent it through our copy editor.

Reviewer 3 Report

Comments and Suggestions for Authors

Manuscript written by Joseph W. Golden  , Steven A. Kwilas and Jay W. Hooper present the  studies devoted to develop the immunotherapy against severe  disease caused by New World (NW) arenaviruses, including Junin virus (JUNV) and Machupo virus (MACV).Current manuscript present a significant progress in  evaluation of developing immunity against NW arenaviruses.  Manuscript is well written, detailed description of assays used in studies are presented clearly and convincible in a paper. Conclusions based on results are valid. Discussion is written and organized very well. Being the virologist I enjoy reading the detailed discussion with good description of previous studies in a  field. 

I have some specific comments: Figure 2 is poorly  organized. There are no proper legends  for  2A, 2B and 2C. , even there is no indication for A,B and C. 

Author Response

Manuscript written by Joseph W. Golden  , Steven A. Kwilas and Jay W. Hooper present the  studies devoted to develop the immunotherapy against severe  disease caused by New World (NW) arenaviruses, including Junin virus (JUNV) and Machupo virus (MACV).Current manuscript present a significant progress in  evaluation of developing immunity against NW arenaviruses.  Manuscript is well written, detailed description of assays used in studies are presented clearly and convincible in a paper. Conclusions based on results are valid. Discussion is written and organized very well. Being the virologist I enjoy reading the detailed discussion with good description of previous studies in a  field. 

I have some specific comments: Figure 2 is poorly  organized. There are no proper legends  for  2A, 2B and 2C. , even there is no indication for A,B and C. 

  • Author response: This has been remedied. Some missing information was added to make the figure more clear. We apologize for our oversight.

Round 2

Reviewer 2 Report

Comments and Suggestions for Authors

I hav no other concerns.